# Mechanisms of Oncolytic Virus-Induced Multi-Modal Cell Death and Therapeutic Prospects

**DOI:** 10.3390/ijms26199770

**Published:** 2025-10-07

**Authors:** Jinzhou Xu, Chenqian Liu, Ye An, Jianxuan Sun, Shaogang Wang, Qidong Xia

**Affiliations:** Department and Institute of Urology, Tongji Hospital, Tongji Medical College, Huazhong University of Science and Technology, Wuhan 430000, China; jason980620@163.com (J.X.); chenqian.liu@utsouthwestern.edu (C.L.); ay121253@163.com (Y.A.); sunjianxuan123@126.com (J.S.)

**Keywords:** oncolytic virus, cell death, cancer therapy, apoptosis, immunity

## Abstract

Cancer is a major challenge to global health, and its incidence rate and mortality are expected to continue to rise in the coming decades. Traditional treatment methods such as surgery, radiotherapy, and chemotherapy have limitations, which has prompted people to explore new treatment strategies. As a promising therapeutic approach, oncolytic viruses can selectively target and lyse tumor cells while avoiding damage to normal tissues. This article systematically reviews the mechanisms by which oncolytic viruses induce various forms of cell death, including apoptosis, autophagy, pyroptosis, necroptosis, and ferroptosis. We explored the direct killing effect of oncolytic viruses and their ability to activate local and systemic antitumor immunity, with a focus on the latest developments in the clinical application of oncolytic viruses, such as the development of novel recombinant viruses. In addition, we also analyzed strategies to enhance the efficacy of oncolytic viruses through gene modification, combination therapy, and targeted delivery systems. A deeper understanding of the multiple mechanisms of action of oncolytic viruses can help us develop more effective and personalized cancer treatment plans. Future research should focus on optimizing oncolytic viruses to overcome tumor drug resistance and improve patient prognosis, making them an important pillar of cancer treatment.

## 1. Introduction

Cancer is one of the major “health killers” in our daily lives. In 2020, the global incidence of cancer reached 19.3 million new cases, with approximately 10 million cancer-related deaths. It is estimated that, by 2040, the number of cancer patients worldwide will increase to 28.4 million, representing a 47% increase compared to 2020 levels [1]. A century ago, cancer was not widely recognized, but with societal development, changes in lifestyle and dietary habits, and advancements in diagnostic techniques, cancer has progressively entered public awareness and has become a feared disease. Cancer remains a major challenge that has yet to be fully overcome, and many researchers dedicate their lives to its study. Traditional cancer treatment methods include surgery, radiotherapy, and chemotherapy. Surgery was the first successful approach for treating cancer and remained the sole treatment option for many common solid tumors for many years. In addition to these conventional methods, advances in cancer research have led to the widespread application of targeted therapy, immunotherapy, gene therapy, and hormone therapy. However, different cancers respond variably to different treatment modalities, highlighting the long road ahead for cancer treatment research.

Cancer patients often hope for miracles, and such hopes are not entirely unfounded. In the 13th century, a Catholic priest suffering from tibial tumor also developed a severe skin infection, and unexpectedly, his tumor regressed spontaneously [2]. Since then, numerous cases of spontaneous tumor regression have been reported, with a common factor among them—acute infection. This observation led scientists to explore the potential of bacterial and viral agents as cancer therapies. A prominent example is Bacillus Calmette–Guérin (BCG), originally developed as a tuberculosis vaccine, which was successfully used to treat bladder cancer in 1976. Since then, it has become the first-line treatment for non-muscle invasive bladder cancer and remains widely used today [3]. Oncolytic viruses, a class of viral agents, have recently emerged as a promising cancer therapy. Compared to conventional chemotherapeutic agents, oncolytic viruses selectively target and destroy tumor cells while sparing normal cells, thus offering a more precise therapeutic approach [4].

Most published reviews on oncolytic viruses focus on their role in inducing antitumor immune responses while ignoring their direct cytotoxic effects. Therefore, this review aims to summarize and discuss the direct oncolytic effects of oncolytic viruses on tumor cells and their ability to induce different forms of cell death, providing insights into future therapeutic applications.

## 2. Oncolytic Virus

Oncolytic viruses can be classified into DNA and RNA viruses based on their nucleic acid composition. DNA oncolytic viruses include adenoviruses, vaccinia viruses, and herpes simplex viruses (HSV), while RNA oncolytic viruses include reoviruses, coxsackieviruses, measles viruses, and Newcastle disease viruses [4]. The virological characteristics of common oncolytic viruses are presented in Table 1.

With advancements in oncolytic virus research, these viruses have transitioned from basic research to clinical applications (Table 2). In 2015, the U.S. Food and Drug Administration (FDA) approved the first oncolytic virus, talimogene laherparepvec (T-VEC), for the treatment of metastatic melanoma. T-VEC is a genetically modified herpes simplex virus type 1 (HSV-1) engineered to express granulocyte-macrophage colony-stimulating factor (GM-CSF) [5]. The development and approval of T-VEC marked a significant milestone in oncolytic virotherapy. Subsequently, in 2021, Japan approved the world’s first oncolytic virus for malignant glioma, G47Δ, which is a third-generation HSV-1 with triple mutations [6].

The antitumor mechanisms of oncolytic viruses can be broadly categorized into two major types: direct oncolysis and activation of antitumor immunity.

Direct Oncolysis: Most oncolytic viruses directly lyse host tumor cells. The effectiveness of this direct cytotoxicity is influenced by viral tropism, viral replication efficiency, and the host tumor cells’ antiviral response [19]. Oncolytic virus tropism is primarily determined by the ability of the virus to recognize specific receptors on the tumor cell surface. For example, measles virus selectively targets tumor cells by binding to SLAM (CD150) and CD46 receptors, while herpes simplex virus (HSV) specifically interacts with HVEM, nectin-1, and nectin-2 receptors [20]. The direct lytic potential of oncolytic viruses also depends on viral strain, dosage, and the susceptibility of tumor cells to different forms of cell death, including apoptosis, autophagy, and pyroptosis [4].

Activation of Local and Systemic Immunity: Upon infection of tumor cells, oncolytic viruses replicate intracellularly and induce the release of damage-associated molecular patterns (DAMPs) and pathogen-associated molecular patterns (PAMPs), which initiate antitumor immune responses [21]. Additionally, oncolytic viruses enhance the expression of major histocompatibility complex (MHC) proteins and costimulatory molecules such as CD40, CD80, and CD86 on dendritic cells (DCs) [22]. DCs present viral or tumor-associated antigens via MHC class I and MHC class II pathways, leading to the activation of CD8+ and CD4+ T cells, respectively. Activated CD4+ T cells secrete cytokines such as IL-12 and IFN-γ, while CD8+ T cells mediate cytotoxicity through perforin–granzyme and Fas-FasL pathways.

## 3. Oncolytic Virus and Apoptosis

Apoptosis is characterized by distinct morphological changes, including cell shrinkage, chromatin condensation, and nuclear fragmentation. Previous studies have suggested that caspase activation is essential for the apoptotic phenotype and that the degree of caspase activation may serve as an indicator of apoptosis progression [23]. Caspase activation generally occurs through three primary pathways—the intrinsic mitochondrial pathway, the extrinsic death receptor pathway, and the intrinsic endoplasmic reticulum (ER) stress pathway—with the first two being the most observed [24].

Oncolytic viruses have been demonstrated to induce apoptosis in tumor cells (Figure 1) [25]. Adenoviruses provide a classic paradigm for virus-induced apoptosis. The adenoviral early protein E1A drives uncontrolled cell cycle progression, which inherently predisposes the cell to apoptosis. To counteract this and ensure successful viral replication, adenoviruses express E1B-19K (a functional homolog of the anti-apoptotic protein Bcl-2) to inhibit the mitochondrial apoptosis pathway, and E1B-55K, which binds to and degrades p53, thereby blocking p53-mediated apoptosis [26]. Oncolytic herpes simplex virus type 1 (oHSV-1) has been shown to trigger apoptosis in multiple myeloma cells via Caspase-3 cleavage [27]. Reovirus reduces Ras palmitoylation, leading to Golgi fragmentation and Ras accumulation at the Golgi, which subsequently enhances viral release and promotes apoptosis [28]. The genomic RNA fragments of Sendai virus selectively induce apoptosis in cancer cells by upregulating tumor necrosis factor-related apoptosis-inducing ligand (TRAIL) and the downstream effectors of the retinoic acid-inducible gene I (RIG-I)/mitochondrial antiviral signaling (MAVS) pathway, such as NOXA [29]. Newcastle disease virus (NDV) increases lysosomal membrane permeability, resulting in the translocation of cathepsins B and D and triggering mitochondria-dependent apoptosis [30]. Furthermore, NDV infection in non-small cell lung cancer cells overexpressing Bcl-xL can lead to enhanced syncytia formation and subsequent massive apoptosis [31]. M1 virus, a naturally occurring oncolytic virus and Getah-like alphavirus strain isolated in Hainan Province, China [32], was first identified and reported for its oncolytic activity in 2014 by Professor Guangmei Yan’s group during their investigations of arboviruses [33]. It has been found to induce sustained and severe ER stress, leading to apoptosis in tumor cells deficient in zinc-finger antiviral proteins [34,35]. Building on this, Lin et al. discovered that knockout of coiled-coil domain-containing protein 6 (CCDC6) enhanced M1 virus replication and increased ER stress-induced apoptosis in both in vitro and in vivo bladder cancer models [36]. Similarly, Cai et al. reported that second mitochondria-derived activator of caspase (SMAC) enhanced M1 virus replication, with viral protein accumulation leading to irreversible ER stress and c-Jun N-terminal kinase (JNK)-mediated tumor cell apoptosis [37].

Recognizing the potential of oncolytic viruses in cancer therapy, researchers have sought to enhance their efficacy through genetic modifications or combination therapies with conventional treatments. Tamura et al. engineered a recombinant oHSV expressing secreted TRAIL (oHSV-TRAIL) and demonstrated that it downregulated the extracellular signal-regulated kinase (ERK)-mitogen-activated protein kinase (MAPK) pathway while upregulating the JNK and p38-MAPK pathways. This modification led to apoptosis in glioblastoma cells via activation of Caspase-8, Caspase-9, and Caspase-3 [38]. Lv et al. developed an oncolytic vaccinia virus (VV) carrying the IL-24 gene (VV-IL-24) and found that high IL-24 expression in lung cancer xenografts reduced STAT3 activity, ultimately inducing apoptosis [39]. In a combination therapy setting, the oncolytic virus OBP-301, when used with radiotherapy, reduced STAT3 phosphorylation and its downstream molecule Bcl-xL, while increasing cleaved Caspase-3 expression. In contrast, radiotherapy alone did not induce significant expression of apoptosis-related molecules [35].

Further research has explored combination therapies with chemotherapeutic agents. For example, in a study combining gemcitabine with an oncolytic adenovirus (YDC002) expressing relaxin for pancreatic cancer treatment, the combination therapy significantly reduced the expression of key extracellular matrix components (collagen, fibronectin, and elastin) and effectively induced apoptosis compared to gemcitabine monotherapy [40]. Additionally, researchers have investigated the combined use of measles virus and mumps virus (MM) for the treatment of acute myeloid leukemia. The results indicated that MM infection induced apoptosis, with Fas-FasL interactions playing a partial role in MM-induced apoptosis. The enhanced oncolytic effect observed in combination therapy was attributed to increased tumor cell apoptosis [41].

It has been reported that oncolytic virus infections can trigger immune responses that may attenuate their antitumor efficacy. To circumvent this issue, researchers have attempted to encapsulate oncolytic viruses to evade immune phagocytosis and enhance targeted delivery to tumor sites [42]. An Iranian study employed adipose-derived mesenchymal stem cells (AD-MSCs) as carriers for the delivery of reovirus, demonstrating that the modified virus significantly upregulated the expression of apoptotic genes, including P21, P53, Bax, Bid, Caspase-8, and Caspase-3, while downregulating anti-apoptotic genes such as Bcl-xL and Bcl-2 in murine colorectal cancer cells [43].

As previously mentioned, M1 virus exerts its antitumor effects via sustained ER stress [35]. However, Li et al. found that this prolonged and severe ER stress does not occur in refractory malignant tumors. Instead, they discovered that activation of the cAMP signaling pathway suppressed the expression of antiviral factors induced by M1 virus in these tumors, thereby restoring M1 virus efficacy and leading to persistent ER stress and apoptosis [44].

## 4. Oncolytic Virus and Autophagy

Autophagy is a cellular homeostasis mechanism that enables cells to recognize and internalize foreign pathogens, subsequently degrading them via autophagosomes and lysosomes in response to cellular stress [45]. It is a double-edged sword, playing both protective and cytotoxic roles. Under normal physiological conditions, autophagy facilitates the clearance of damaged organelles to generate energy [46]. During tumor progression, cancer cells can exploit autophagy to maintain their pathological state [47]. The primary mechanisms and regulatory factors involved in tumor-associated autophagy include the PI3K/AKT pathway, the LKB1-AMPK pathway, and p53 [48]. 

The balance between autophagy and cell death in cancer cells infected with oncolytic viruses is virus-dependent [49]. Oncolytic virus infection can induce autophagy-mediated tumor cell death. Beclin-1 plays a crucial role in autophagy, serving as a key component of the major catabolic pathway responsible for the degradation of macromolecules and damaged organelles [50]. The adenoviral E1A protein has been identified as a potent inducer of autophagy, capable of activating this process by upregulating autophagy-related gene transcription, inhibiting the mTORC1 signaling pathway, and inducing endoplasmic reticulum stress [51,52]. Vesicular stomatitis virus (VSV) has been shown to induce autophagy-mediated breast cancer cell death by increasing Beclin-1 expression [53]. A novel oncolytic vaccinia virus expressing Beclin-1 (OVV-BECN1) was found to upregulate SIRT-1, leading to LC3 deacetylation and its translocation from the nucleus to the cytoplasm, thereby inducing autophagy-mediated cell death [54].

Tong et al. developed a chimeric oncolytic adenovirus carrying Beclin-1 (SG511-BECN) and investigated its oncolytic effects on tumors. Their findings demonstrated that SG511-BECN treatment induced strong autophagy-mediated cell death in leukemia cells, prolonged survival, and reduced tumor volume in a murine leukemia model [55]. Moreover, the combination of SG511-BECN with the chemotherapeutic agent doxorubicin enhanced viral infection efficiency [56]. Oncolytic virus OBP-301 was shown to increase the expression of the autophagy-related marker LC3-II in neuroblastoma cells [57]. Another study reported that OBP-301 upregulated miR-7, which inhibited cell viability and induced autophagy-mediated cell death by suppressing EGFR expression [58]. FK866, a nicotinamide phosphoribosyl transferase inhibitor, was found to enhance autophagy-mediated cell death in multiple myeloma cells when combined with reovirus, primarily by reducing glucose flux into the tricarboxylic acid (TCA) cycle in KMS12 cells [59]. Additionally, an engineered bacterial outer membrane vesicle-based oncolytic virus delivery system has been shown to enhance antitumor activity by promoting autophagy [59].

However, autophagy induced by oncolytic virus infection may also provide protective effects for tumor cells (Figure 2). M1 virus was found to induce autophagy in glioma cells, which limited its antitumor activity by degrading viral proteins. During viral replication, the unfolded protein response (UPR) may restrict the replication of oncolytic virus proteins. The accumulation of misfolded proteins in the endoplasmic reticulum lumen activates the IRE1α-XBP1 pathway, and studies have shown that inhibition of IRE1α can block M1 virus-induced autophagy [60].

## 5. Oncolytic Virus and Pyroptosis

Pyroptosis is a form of programmed cell death driven by inflammasome activation and caspase activation. It is characterized by distinct morphological changes, including cell swelling, membrane blebbing, and eventual cell lysis. Inflammasome-associated caspases, such as Caspase-1, Caspase-4, Caspase-5, and Caspase-11 (in mice), play critical roles in pyroptosis. Additionally, apoptosis-related caspases, including Caspase-3 and Caspase-8, are also essential in this process [61].

Previous studies have shown that pyroptosis is typically classified into the "canonical" pathway mediated by Caspase-1 and the "non-canonical" pathway mediated by Caspase-4/5/11. Both pathways involve the cleavage of gasdermin D (GSDMD), generating an active N-terminal fragment that perforates the cell membrane and triggers pyroptosis [62]. Furthermore, apoptosis-related Caspase-3 and Caspase-8 can act upstream of gasdermin E (GSDME), converting apoptosis into pyroptosis [63].

Several oncolytic viruses have been demonstrated to induce pyroptosis via these pathways (Figure 3). Oncolytic parapoxvirus and its recombinant therapeutic derivatives can induce pyroptosis in tumor cells via GSDME activation. This effect is particularly prominent in tumor cells with low GSDME expression, where the virus stabilizes GSDME by reducing its ubiquitination, thereby predisposing cells to pyroptosis [64]. Coxsackievirus B3 has also been reported to induce pyroptosis in colorectal cancer cells by activating Caspase-3, leading to GSDME cleavage rather than GSDMD cleavage [65]. The recombinant measles virus vaccine strain rMV-Hu191 has been shown to trigger mitochondrial dysfunction via BAK or Bax, inducing pyroptosis through the Caspase-3/GSDME pathway [59]. Genetically modified OVs and their combination with novel nanoparticle drugs are being explored to more effectively induce pyroptosis for enhanced anticancer therapy [66,67]. Addsitionally, the Zika virus protease NS2B3 can directly cleave GSDMD, thereby inducing caspase-dependent pyroptosis [68].

## 6. Oncolytic Viruses and Other Forms of Cell Death

Necroptosis is a regulated form of necrosis that relies on RIPK1- and RIPK3-mediated phosphorylation of MLKL protein [69]. RIPK3 is the key to necrotic apoptosis and can phosphorylate MLKL. Phosphorylated MLKL (p-MLKL) oligomerizes to form activated necrosomes, which transfer to the cell membrane and ultimately lead to cell death [70]. Zhang et al. found that M1 virus can induce ER stress-dependent non-apoptotic death in triple-negative breast cancer cells. Further research confirms that, after M1 virus infection, the Ser358 site of MLKL undergoes phosphorylation and transfers from the nucleus to the cell membrane, indicating that M1 virus induces necrotic apoptosis. In addition, they also found that doxorubicin can promote the replication of M1 virus through the Gas/STAT1/interferon related pathway [71]. Chen et al.’s study showed that the combination of stereotactic radiotherapy and oncolytic poxvirus can induce necrotic apoptosis of tumor cells and activate macrophages by releasing damage associated molecular patterns (DAMPs), thereby enhancing antitumor effects [72].

NETosis is an inflammatory death mode of neutrophils, characterized by chromatin depolymerization. Neutrophil extracellular traps (NETs) are composed of depolymerized chromatin and various post-translational modified proteins [73]. In the tumor microenvironment, tumor cells can directly induce NET formation, thereby capturing circulating tumor cells and promoting their adhesion, migration, and invasion [74]. In addition, viral infection can also activate neutrophils to produce NETs [54]. Dai et al. found that oncolytic herpes simplex virus induces NET formation by upregulating IGF2BP3 in glioblastoma, thereby hindering the therapeutic effect of oncolytic virus on malignant tumors. Subsequent studies have shown that using BET inhibitors to block IGF2BP3-mediated NETosis can enhance the killing effect of oncolytic herpes simplex virus [75].

Ferroptosis is caused by excessive accumulation of lipid peroxides due to metabolic disorders within cells. Iron death is closely related to intracellular iron metabolism and lipid homeostasis, and is regulated by various pathways such as redox balance, iron metabolism, mitochondrial function, and amino acid, lipid, and glucose metabolism [61]. Xie et al. found that Newcastle disease virus can induce ferroptosis by activating p53 and inhibiting the Xc^−^ system (a cystine/glutamate antiporter that maintains redox balance) [76]. At the same time, the virus can further promote iron death by releasing ferrous ions, enhancing the Fenton reaction, and inducing ferritin autophagy [77].

## 7. Summary and Outlook

As an emerging cancer treatment method, oncolytic viruses are receiving increasing attention from researchers. They have shown broad application prospects by directly lysing cells, inducing various death modes, and activating local and systemic immunity. This article focuses on the research progress and specific mechanisms of cell death induced by oncolytic viruses, including apoptosis, autophagy, pyroptosis, necroptosis, and ferroptosis. These mechanisms not only contribute to a deeper understanding of the characteristics of oncolytic viruses, but also provide a basis for developing more effective tumor therapies.

Notably, most OVs do not operate through a single death pathway; instead, they enhance antitumor efficacy and reduce the likelihood of resistance by synergistically activating multiple cell death modalities. Future research should focus on improving OV targeting, delivery efficiency, and immune activation, as well as exploring combination strategies with other treatment modalities.

With the deepening of research, oncolytic virus therapy will be safer and more widely applied, bringing new hope to cancer patients.

## Figures and Tables

**Figure 1 ijms-26-09770-f001:**
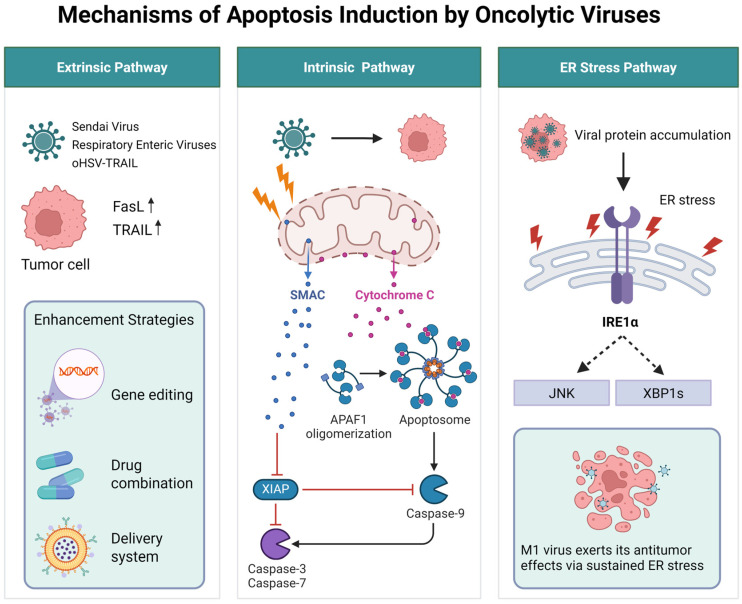
The mechanisms of Oncolytic Virus-Induced Apoptosis.

**Figure 2 ijms-26-09770-f002:**
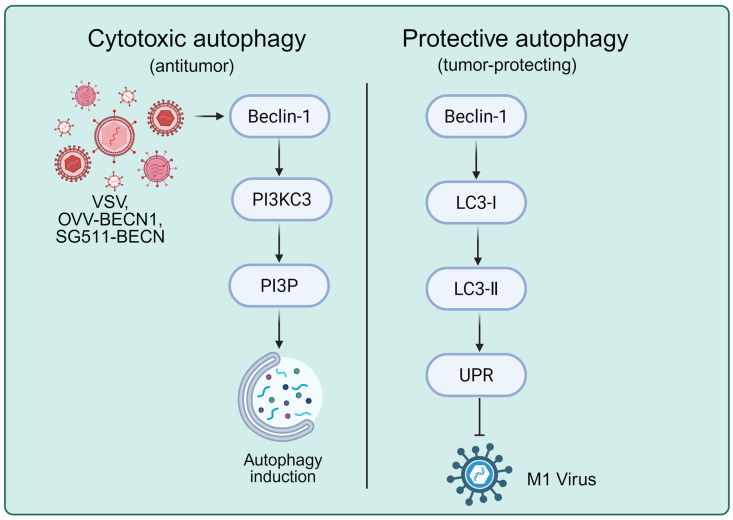
The mechanisms of Oncolytic Virus-Induced Autophagy.

**Figure 3 ijms-26-09770-f003:**
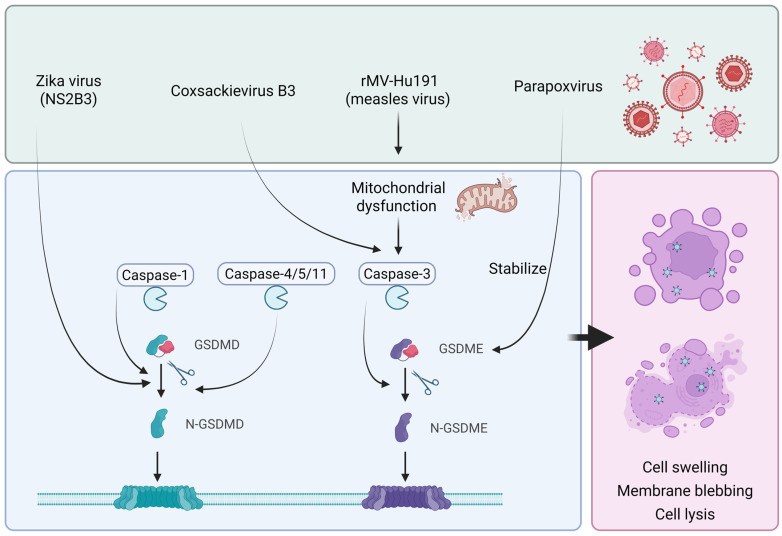
The mechanisms of Oncolytic Virus-Induced Pyroptosis.

**Table 1 ijms-26-09770-t001:** Common types and basic characteristics of oncolytic viruses.

Name	Classification	Size (nm)	Capsid Coating	Extracellular Receptor
Adenovirus	dsDNA	70–90	No	CAR
Vaccinia virus	dsDNA	70–100	Yes	Unknown
Herpesvirus	dsDNA	200	Yes	HVEM, nectin1, nectin2
Reovirus	dsRNA	75	No	Unknown
Coxsackievirus	ssRNA	28	No	CAR, DAF, ICAM-1
Measles virus	ss(-)RNA	100–200	Yes	CD46, CD150, SLAM
Newcastle disease virus	ss(-)RNA	100–400	Yes	sialic acid receptors

Note: CAR, Coxsackievirus and Adenovirus Receptor; HVEM, herpesvirus entry mediator; DAF, decay-accelerating factor; ICAM1, intercellular adhesion molecule-1; SLAM, signaling lymphocyte-activation molecule.

**Table 2 ijms-26-09770-t002:** Representative oncolytic viruses in clinical development and application.

Viral Name	Viral Type	Interventions	Indication	Clinical Phase	Identifier.	Clinical Responses
T-VEC	HSV	T-VEC	IIIB to IV melanoma	Phase III	NCT00769704 [7]	Median OS was 23.3 months (19.5–29.6 months) with T-VEC and 18.9 months (16.0–23.7 months) with GM-CSF
T-VEC	HSV	T-VEC plus NAC	nonmetastatic triple-negative breast cancer	Phase II	NCT02779855 [8]	RCB0 rate = 45.9% and RCB0–1 descriptive rate = 65%.
T-VEC	HSV	T-VEC plus pembrolizumab	Advanced Melanoma	Phase III	NCT02263508 [9]	T-VEC-pembrolizumab did not significantly improve PFS or OS compared with placebo-pembrolizumab.
G47∆	HSV	G47∆	residual or recurrent, supratentorial glioblastoma	Phase II	UMIN000015995 [6]	1-yr survival rate: 84.2% (60.4–96.6)
CAN-3110	HSV	CAN-3110	recurrent glioblastoma	Phase I	NCT03152318 [10]	median OS: 11.6 months (7.8–14.9 months)
NDV-GT	NDV	NDV-GT	advanced malignant solid tumors	Phase I	ChiCTR2000031980 [11]	disease control rate: 90%
VG161	HSV	VG161	refractory hepatocellular carcinoma	Phase I	NCT04806464 [12]	median PFS: 2.9 months (1.81–3.70) and OS: 12.4 months (7.10–20.10)
DNX-2401	AD	DNX-2401	diffuse intrinsic pontine glioma	Phase I	NCT03178032 [13]	median OS 17.8 months
DNX-2401	AD	DNX-2401 plus pembrolizumab	recurrent glioblastoma	phase I/II	NCT02798406 [14]	Median OS: 12.5 months (10.7–13.5 months)
JX-594	VV	JX-594	Melanoma	phase I/II	NCT00429312 [15]	DOR and PFS were not assessable since most patients went off study within 6 weeks
JX-594	VV	JX-594	Neoplasms, liver	Phase I	NCT00629759 [16]	Median OS: 9 months
GL-ONC1	VV	GL-ONC1	Peritoneal Carcinomatosis	Phase I	NCT01443260 [17]	GL-ONC1 was well tolerated when administered into the peritoneal cavity of patients with advanced stage peritoneal carcinomatosis
GL-ONC1	VV	GL-ONC1	Locoregionally Advanced Head and Neck Carcinoma	Phase I	NCT01584284 [18]	1-year (2-year) PFS and OS were 74.4% (64.1%) and 84.6% (69.2%)

Abbreviations: HSV, Herpes Simplex Virus; NAC, neoadjuvant chemotherapy; RCB, residual cancer burden index; NDV, Newcastle disease viruses; AD, adenovirus; VV, vaccinia virus; DOR, Duration of Response; PFS, progression-free survival; OS, overall survival.

## Data Availability

No new data were created or analyzed in this study. Data sharing is not applicable to this article.

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
