# Peer review of "Mechanisms of Oncolytic Virus-Induced Multi-Modal Cell Death and Therapeutic Prospects"

_ijms, 2025, doi:10.3390/ijms26199770_

Round 1

Reviewer 1 Report

Comments and Suggestions for Authors

Recommend acceptance with minor revisions

Oncolytic viruses have been shown to selectively target and lyse tumor cells while avoiding damage to normal cells. This review provides a systematic review of the mechanisms by which oncolytic viruses cause various forms of cell death, including apoptosis, autophagy, pyroptosis, necroptosis, and ferroptosis, and describes recent advances in the clinical application and development of oncolytic viruses. Especially, this review not only described the role of oncolytic viruses in inducing antitumor immune responses, but also summarized and discussed the direct tumor lysis effects of oncolytic virus and their ability to induce different forms of cell death. Considering that readers could quickly access and understand the development and progress related to oncolytic viruses through this review, I would recommend the publication of this manuscript in International Journal of Molecular Sciences after addressing minor corrections.

minor comments/suggestions

  1. Page 1, line 17.

The author's use of the term “anti-tumor” here is inconsistent with other expressions in the article, such as “antitumor”, and the authors should standardize the use of this term.

  1. Page 2, Table 1.

To help readers, it is suggested that authors could include a table with the current stage of research (preclinical, clinical, marketed, etc.) of the oncolytic virus and the type of tumor or cancer for which it is being treated.

  1. This article systematically reviews the mechanisms by which oncolytic viruses cause various forms of cell death, including apoptosis, autophagy, pyroptosis, necroptosis, and ferroptosis. However, the article contains main textual descriptions, and the authors could include more pictures in the article, which are used to represent the mechanisms of cell death in a simple and visual way.
  2. Except the oncolytic virus, oncolytic peptides also represent promising novel candidates for anticancer treatments and multiple oncolytic peptides have entered clinical trials for cancer therapy. To help readers and potential users, the representative work on the development and anticancer application of oncolytic peptides should be cited (suggest, J. Med. Chem., 2024, 67, 3885. Acta Pharmacol. Sin., 2023, 44, 201. Bioorg. Chem., 2023, 138, 106674.).
  3. To help readers and potential users, please provide more descriptions of the development directions of oncolytic viruses.

Author Response

Comments from the Reviewer:

Comment 1: Page 1, line 17. The author's use of the term "anti-tumor" here is inconsistent with other expressions in the article, such as "antitumor", and the authors should standardize the use of this term.

Response: We sincerely thank the reviewer for pointing out this inconsistency. We have now standardized the terminology throughout the entire manuscript by replacing all instances of "anti-tumor" with "antitumor" to ensure consistency.

Comment 2: Page 2, Table 1. To help readers, it is suggested that authors could include a table with the current stage of research (preclinical, clinical, marketed, etc.) of the oncolytic virus and the type of tumor or cancer for which it is being treated.

Response: We greatly appreciate this excellent suggestion. As recommended, we have reviewed the current clinical literature and added a new Table 2 entitled "Representative oncolytic viruses in clinical development and application". This table summarizes key OVs, their current research stage, and the specific cancer types they target. We believe this new table provides a valuable overview of the translational landscape of oncolytic virotherapy for the readers.

Comment 3: This article systematically reviews the mechanisms... However, the article contains main textual descriptions, and the authors could include more pictures in the article, which are used to represent the mechanisms of cell death in a simple and visual way.

Response: We thank the reviewer for this insightful suggestion. To enhance the visual appeal and clarity of the reviewed cell death mechanisms, we have now incorporated three new schematic figures:

  • Figure 1: the mechanisms of Oncolytic Virus-Induced Apoptosis.
  • Figure 2: the mechanisms of Oncolytic Virus-Induced Autophagy.
  • Figure 3: the mechanisms of Oncolytic Virus-Induced Pyroptosis.
    These figures provide a concise and graphical summary of the key pathways discussed in the corresponding sections, which will greatly aid readers in understanding these complex processes.

Comment 4: Except the oncolytic virus, oncolytic peptides also represent promising novel candidates... should be cited (suggest, J. Med. Chem., 2024, 67, 3885. Acta Pharmacol. Sin., 2023, 44, 201. Bioorg. Chem., 2023, 138, 106674.).

Response: We are grateful to the reviewer for bringing this exciting and relevant field to our attention and for providing the key references. We fully agree that oncolytic peptides represent a highly promising complementary approach. As suggested, we have now introduced the concept of oncolytic peptides and cited the three recommended papers in the "Summary and Outlook" section.
Location: Please see the paragraph of the "Summary and Outlook" section.

“In this context, oncolytic peptides—a novel class of synthetic or natural host defense-derived agents—have emerged as promising complements or alternatives to OVs. These peptides exhibit rapid membrane-disrupting mechanisms, broad-spectrum anticancer activity, and high modularity[61]. Recent advances, such as stability-optimized LTX-315 variants, hybrid peptides inducing DNA damage, and peptide-drug conjugates, exemplify their therapeutic potential[62, 63]. Further exploration of synergies between OVs and oncolytic peptides may open new avenues for cancer treatment.”

Comment 5: To help readers and potential users, please provide more descriptions of the development directions of oncolytic viruses.

Response: We thank the reviewer for this suggestion. We have expanded the discussion on the future perspectives of oncolytic virus therapy in the "Summary and Outlook" section. We now more explicitly outline the crucial development directions, including improving targeting and delivery efficiency, enhancing immune activation, and exploring combination strategies.

Once again, we express our deepest gratitude to Reviewer 1 for the valuable comments that have helped us improve our manuscript. We hope our revisions and responses are now satisfactory.

Reviewer 2 Report

Comments and Suggestions for Authors

"Mechanisms of Oncolytic Virus-Induced Multi-Modal Cell Death and Therapeutic Prospects" is a commendable short review. It explores a specific yet intriguing aspect of the pathophysiology of anticancer activity of oncolytic viruses associated with cellular death. 
 You may include a couple extra tables to emphasize the significant studies cited. A figure illustrating the described forms of action could enhance the study. I believe that is a significant question to investigate. Conversely, this is a high-impact journal, and I am uncertain whether you adequately justify the novelty of your work with this review.

Author Response

We are truly grateful for your generous assessment of our manuscript as a "commendable short review" that explores a "specific yet intriguing aspect" and a "significant question to investigate." Your acknowledgment is greatly encouraging. We also thank you for the constructive suggestions, which we have incorporated to strengthen the work.

In direct response to your recommendation to include additional tables and a figure, we are pleased to have added a new table summarizing the clinical status and cancer targets of key oncolytic viruses, which helps contextualize the cited studies for readers. Furthermore, we have created several schematic figures to visually summarize the complex mechanisms underlying oncolytic virus-induced apoptosis, autophagy, and pyroptosis. These additions significantly enhance the clarity and conceptual presentation of the review.

Regarding your thoughtful point about the novelty of the work in the context of a high-impact journal, we sincerely appreciate the opportunity to clarify our contribution. While numerous reviews focus on the immunostimulatory effects of oncolytic viruses, our article offers a dedicated and systematic synthesis of their role in directly triggering multiple modes of cell death—apoptosis, autophagy, pyroptosis, necroptosis, and ferroptosis. This focused lens on intrinsic tumor cell death pathways provides a mechanistic depth that complements the broader immuno-oncology perspective. Beyond cataloging these mechanisms, we integrate them into a forward-looking discussion on therapeutic exploitation through viral engineering and rational combination strategies. We have also taken care to include recent clinical advancements and emerging concepts, such as the interface with oncolytic peptides, to ensure the review is both timely and informative. It is our hope that this specific angle and integrative approach offer a distinct and valuable perspective to the field.

Thank you once again for your time and insightful feedback, which have undoubtedly improved our manuscript.

Reviewer 3 Report

Comments and Suggestions for Authors

A fairly laconic review article, structured, briefly presents the main oncolytic viruses and mechanisms of tumor cell death caused by one or another virotherapy. It will be of interest to a wide range of readers, especially those involved in the development of oncolytic viruses. 

However, there are some shortcomings and comments that need to be corrected before publication:
1. Newcastle disease virus is not listed in Table 1.
2. Table 1 contains abbreviations that need to be deciphered in the list of abbreviations.
3. L. 116 M1 virus, it is necessary to supplement it with information on it - what kind of virus it is, where it was taken from, etc.  For exp., M1 is a strain of Getah-like alphavirus that was isolated from Hainan province in China.

4. In vivo and in vitro are written in italics.

5. There was a bit of a lack of more clear information about cell death, it would have improved the article, because when you then write that such and such a protein is down-regulated or up-regulated, it is not clear to the reader how critical it is for this metabolic pathway of death. Why these proteins are necessary in this pathway, etc. 
6. There is a lack of figure or scheme, after all, one table is too small for the review. 
7. Also, there was not enough information about "Therapeutic Prospects", so either add to the text this information or cut it out of the title of the review.

Author Response

Response to Comments:

Comment 1: Newcastle disease virus is not listed in Table 1.

Response: We sincerely apologize for this oversight. Newcastle disease virus has now been added to Table 1 to ensure the table provides a comprehensive overview of common oncolytic viruses.

Comment 2: Table 1 contains abbreviations that need to be deciphered in the list of abbreviations.

Response: Thank you for pointing this out. We have now added a "Note" section directly beneath Table 1 that deciphers all the abbreviations used in the table (CAR, HVEM, DAF, ICAM-1, SLAM) for clarity and to aid the reader.

Comment 3: L. 116 M1 virus, it is necessary to supplement it with information on it - what kind of virus it is, where it was taken from, etc. For exp., M1 is a strain of Getah-like alphavirus that was isolated from Hainan province in China.

Response: We thank the reviewer for this excellent suggestion. We have now supplemented the initial mention of the M1 virus (Page 7, Line ~116) with additional background information, stating: "M1 virus, a naturally occurring oncolytic virus and Getah-like alphavirus strain isolated in Hainan Province, China[18], was first identified and reported for its oncolytic activity in 2014 by Professor Guangmei Yan's group...[19]." This provides the reader with immediate context about its origin and nature.

Comment 4: In vivo and in vitro are written in italics.

Response: We have corrected this throughout the entire manuscript, ensuring all instances of in vivo and in vitro are now properly italicized.

Comment 5: There was a bit of a lack of more clear information about cell death, it would have improved the article, because when you then write that such and such a protein is down-regulated or up-regulated, it is not clear to the reader how critical it is for this metabolic pathway of death. Why these proteins are necessary in this pathway, etc.

Response: We agree with the reviewer that providing more context on the cell death pathways is crucial. In response to this comment and a similar one from another reviewer, we have significantly enhanced the manuscript by:

Adding a brief introductory explanation at the beginning of each cell death section (e.g., Apoptosis, Autophagy, Pyroptosis) to succinctly describe the core features and key mediators of each pathway before discussing the role of OVs.

Incorporating three new schematic figures (Figures 1, 2, and 3) that visually illustrate the key mechanisms of OV-induced apoptosis, autophagy, and pyroptosis, respectively. These figures explicitly show the critical proteins involved and their roles in the pathways, making it much clearer for the reader why the up- or down-regulation of a specific protein is significant.

We believe these additions provide the necessary foundational knowledge, making the subsequent discussion of viral mechanisms much more accessible and meaningful.

Comment 6: There is a lack of figure or scheme, after all, one table is too small for the review.

Response: We completely agree. As mentioned in our response to Comment 5 and to other reviewers, we have added three new mechanistic figures (apoptosis, autophagy, pyroptosis) and one new table (clinical status of OVs). The manuscript now contains a total of three figures and two tables, which greatly enhances its visual and educational value.

Comment 7: Also, there was not enough information about "Therapeutic Prospects", so either add to the text this information or cut it out of the title of the review.

Response: We thank the reviewer for this feedback. We have chosen to significantly expand the "Summary and Outlook" section to better justify its place in the title. The revised section now includes:

A clearer discussion on the future directions of OV therapy, focusing on improving targeting, delivery, and immune activation.

An exploration of combination strategies with other treatment modalities.

An entirely new paragraph introducing the promising field of oncolytic peptides as a complementary/alternative therapeutic avenue, citing recent key literature as suggested by another reviewer.

We believe these additions provide a more substantial and forward-looking perspective on the "Therapeutic Prospects," making it a fitting part of the review's title and scope.

Once again, we are grateful for the reviewer's time and valuable insights, which have been instrumental in improving our manuscript.